# Interleukin-17 Family Cytokines in Metabolic Disorders and Cancer

**DOI:** 10.3390/genes13091643

**Published:** 2022-09-13

**Authors:** Eileen Victoria Meehan, Kepeng Wang

**Affiliations:** Department of Immunology, School of Medicine, University of Connecticut Health Center, 263 Farmington Ave, Farmington, CT 06030, USA

**Keywords:** interleukin-17, metabolic disorders, inflammation, cancer

## Abstract

Interleukin-17 (IL-17) family cytokines are potent drivers of inflammatory responses. Although IL-17 was originally identified as a cytokine that induces protective effects against bacterial and fungal infections, IL-17 can also promote chronic inflammation in a number of autoimmune diseases. Research in the last decade has also elucidated critical roles of IL-17 during cancer development and treatment. Intriguingly, IL-17 seems to play a role in the risk of cancers that are associated with metabolic disorders. In this review, we summarize our current knowledge on the biochemical basis of IL-17 signaling, IL-17′s involvement in cancers and metabolic disorders, and postulate how IL-17 family cytokines may serve as a bridge between these two types of diseases.

## 1. IL-17 Family Cytokines and Their Receptors

Initially detected in murine T cell hybridoma clones and identified as cytotoxic T lymphocyte-associated antigen 8 (CTLA-8) in 1993, interleukin-17, also known as interleukin-17A (IL-17A), was the first member of the IL-17 family of cytokines to be discovered [1,2]. In mammals, there are currently six identified members of the IL-17 family: IL-17A, IL-17B, IL-17C, IL-17D, IL-17E, and IL-17F [1,2,3,4,5,6,7,8]. Each member of the family shares some degree of homology in amino acid sequence with IL-17A, ranging from just under 20% to around 55% in similarity [9,10]. IL-17F is located next to IL-17A on human chromosome 6 and mouse chromosome 1 but is transcribed independently. IL-17F is IL-17A’s closest homolog— it has an amino acid sequence similarity of around 40 or 55% depending on the IL-17F isoform [9,10,11,12]. The next closest homolog is IL-17B, which shares around 29% amino acid sequence similarity with IL-17A, followed by IL-17D at 25%, IL-17C at 23%, and the most divergent homolog IL-17E (also called IL-25) at roughly 17% [7,8,12]. The IL-17 family of cytokines are homodimeric glycoproteins, or heterodimeric in the case of the IL-17A/F cytokine, and these dimers range from around 35 to 50 kDa in size [4,10,12]. The IL-17 family of cytokines share a conserved C-terminus, which contains four cysteine and two serine residues, which are necessary to form the cysteine knot fold observed in the crystal structures of IL-17A and IL-17F [4,9,10,12,13]. The IL-17 family of cytokines diverge in their N-terminal segments [4,10,12]. What is known about the structural aspects of these molecules has been extensively described in reviews by Zhang et al. (2011) and Liu (2019) [14,15].

The six members of the IL-17 family also differ in receptor binding preferences. There are five currently identified members in the IL-17 receptor family: IL-17RA, IL-17RB, IL-17RC, IL-17RD, and IL-17RE [16]. Each receptor polypeptide shares sequence homology with IL-17RA, and many of the genes encoding the IL-17 receptors are clustered on human chromosome 3 or on mouse chromosomes 6 and 14 [14,16,17]. Except for IL-17RA, IL-17 receptors can undergo alternative splicing, sometimes leading to soluble secreted proteins that inhibit IL-17 signaling [12,17,18]. The IL-17 receptors are unique, in that, given the current structural knowledge, they do not fit into any of the six major families of cytokine receptors [14]. The receptors are single-pass, type I transmembrane domain-containing receptors, which share conserved structural motifs, such as a fibronectin III-like domain in the extracellular region and a SEF/IL-17R (SEFIR) domain in the intracellular region [5,16,19]. IL-17 receptors bind to IL-17 family cytokines as homo- or heterodimers and transduce signal through adaptor protein NF-κB activator 1 (ACT1) and E3 ubiquitin ligase tumor necrosis factor receptor-associated factor protein (TRAF) TRAF6, which leads to the activation of nuclear factor kappa B (NF-κB), mitogen-activated protein kinases (MAPK), and CCAAT/enhancer binding protein (C/EBP) pathways [20]. Detailed information on the signaling cascade of IL-17 family cytokines will be introduced in later sections.

IL-17RA was first identified as the receptor for IL-17A, but it also binds to IL-17F and the IL-17A/F heterodimer [2,9,16,21,22]. Alone, IL-17RA has a high affinity for IL-17A, a much weaker affinity for IL-17F, and an intermediate affinity for IL-17A/F, as well as an even weaker affinity for the other four cytokines in the IL-17 family [4,20,21,23]. The IL-17RC and IL-17RA heterodimer is the major receptor complex that mediates IL-17A and F signaling. IL-17RA has been shown to bind asymmetrically with IL-17A and IL-17F in a one receptor chain to one dimer manner (1:2 stoichiometry), whereas the IL-17A/F heterodimer binds to IL-17RA in a binding stoichiometry of one receptor chain per (hetero)dimer (1:1 stoichiometry) [4,9,20,22]. IL-17RA binds to either the ‘A’ face or ‘F’ face of the IL-17A/F heterodimer with similar affinity. While IL-17RA undergoes conformational changes when binding to the ‘F’ face, its structure is not altered upon binding to the ‘A’ face, as induced-fit changes to relieve steric hindrance would be needed [21,22].

IL-17RB (IL-17Rh1) has affinities for both IL-17B and IL-17E but exhibits a stronger affinity for IL-17E [3,6,8]. When binding IL-17B, IL-17RB forms a homodimer, whereas it forms a heteromeric receptor complex with IL-17RA to bind IL-17E [6,8,16,24,25,26].

IL-17RC exists in multiple splice forms. When forming a heterodimeric complex with IL-17RA, IL-17RC binds to IL-17A, IL-17F, and IL-17A/F heterodimer [21,23,27,28,29]. Both IL-17RC and IL-17RA can independently interact with IL-17A or IL-17F, but the complex of both receptors is required to carry out the biological activity of the cytokines, and the binding of a cytokine to one receptor encourages preference for the second receptor-binding site [20]. Recently, it has been shown that IL-17F forms a complex with homodimeric IL-17RC in solution [27]. This may represent how IL-17RC, in its alternatively spliced soluble form, may inhibit IL-17 signaling in the extracellular space [16,22,27]. The mechanism and function of the potential IL-17RC-only transduction of IL-17 signaling remains to be explored and may provide further insight into the development of therapeutic agents for autoimmune diseases, such as psoriasis, where a blockade of IL-17A and IL-17RA showed significant efficacy. From the structure data, we now know that IL-17RC binds to the IL-17A/F heterodimer on either the ‘A’ face or ‘F’ face equally, potentially inferring the formation of two topologically distinct binary complexes [22,27].

IL-17RD (Sef) forms a heterodimer with IL-17RA to selectively bind to IL-17A. It has also been shown to colocalize with IL-17RB [30,31,32]. IL-17RD shares overlapping functions with IL-17RC, such as mediating IL-17A signaling, albeit binding IL-17A at around 20-fold lower affinity than IL-17RC [30]. IL-17RE associates with IL-17RA to bind homodimeric IL-17C [33]. To date, no additional members of the IL-17 family of cytokines have been found to bind to IL-17RE. In addition to the canonical IL-17 receptors, CD93 acts as a receptor of IL-17D and promotes the production of IL-22 by type 3 innate lymphoid cells (iLC3) [34,35].

Overall, the slight differences in IL-17 cytokines and IL-17 receptor structures alter their function. Knowing these differences and their impact on biology will help when identifying therapeutic targets. However, understanding other factors, including how other cytokines affect IL-17-producing cells, is important for understanding IL-17′s role in diseases.

## 2. Multiple Cytokine Pathways Guide the Differentiation of IL-17-Producing Cells

While the inflammatory effects of IL-17 family cytokines are observed as a product of their downstream effects on target cells and induction of their target genes, the upstream pathways have a profound impact on the ability of cells to produce these cytokines and must be considered when trying to understand cellular immunity. Important pathways that have become a focus around the IL-17 family of cytokines include the IL-6/transforming growth factor (TGF) β, IL-23/T helper cell (Th) 17, and IL-1β pathways.

A stimulation of naïve CD4+ T cells with cytokines IL-6 and TGFβ in vitro produces a distinct class of T helper cells, Th17 (Th_IL-17_) cells, and results in a regulatory and protective phenotypic outcome when compared with the phenotypes produced from the IL-23/Th17 axis and the IL-1β axis, which is discussed below [36,37]. Stimulation of Th17 cells with TGFβ and IL-6 results in the production of IL-17A, and in some cases, IL-10. IL-10 mitigates the inflammatory activities of Th17 cells [38,39]. This pathway was initially observed in naïve T cells harboring an activated forkhead box p3 (*Foxp3*) locus, where IL-6 inhibited the induction of Tregs by TGFβ, resulting in their conversion into IL-17-producing cells [39]. Since its initial discovery, IL-6 has been found to be a potent activator of signal transducer and activator of transcription 3 (STAT3), which subsequently suppresses the differentiation of these naïve CD4^+^ T cells into Tregs by regulating the expression of transcription factor retinoic acid receptor-related orphan receptor-γt (RORγt), promoting differentiation into the Th17 subset and subsequent expression of Th17 genes [37,39,40].

The discovery of IL-23, which differentiates naïve CD4^+^ T cells into Th17 cells, expanded our understanding of cellular immunity beyond a dichotomous Th1- and Th2-centric view [37,41,42]. IL-23, which consists of two subunits—p19 and p40, the latter of which it shares with IL-12—is a critical facilitator in promoting inflammation through the induction of Th17 cells that produce IL-17 and additional inflammatory mediators [42,43,44]. Differentiation of CD4^+^ T cells into the inflammatory Th17 subset occurs when antigen-engaging T cells are simultaneously stimulated by IL-6 and TGF-β [39]. Subsequently, IL-23 activates STAT3, further inducing transcription factor RORγt, which is designated as the master transcription factor in IL-23-responsive cells [37,45,46,47,48,49,50,51]. Activation of RORγt results in the production of proinflammatory mediators, such as IL-17A, IL-17F, IL-6, IL-22, and TNFα, which are implicated in the development of multiple inflammatory diseases [42,43,44,51].

In certain inflammatory diseases, IL-23 has been found to promote RORγt expression and reduce Foxp3 expression in regulatory T cells (Tregs), resulting in ‘plastic’ Tregs that produce IL-17A [52,53]. Notably, IL-23 requires additional factors to promote Th17 differentiation, as it is primarily a maintenance factor for Th17 cells and, when acting alone, does not induce differentiation of the Th17 subset [37,42,54,55]. IL-23 functions to expand Th17 cells as well as maintain the cell population by upregulating IL-6, IL-1β, and TNFα through a positive feedback loop [42,53,55]. Th17 differentiation by IL-23 is inhibited by IFN-γ and IL-4 [42].

TGFβ and IL-6 have been predominantly implicated in Th17 differentiation in murine cells, whereas IL-1β has been identified as an important cytokine in human Th17 differentiation [36,37,56]. IL-1β functions alone or together with cytokines, such as IL-23 and IL-6, and regulates the expression of transcription factors interferon regulatory factor 4 (IRF4) and RORγt [36,37,42,55,56,57,58,59,60]. IL-1β promotes the excision of exon 7 of *Foxp3* through alternative splicing, skewing differentiation of naïve T cells toward a Th17 phenotype and away from a Treg phenotype [56]. On their own, neither cytokine is sufficient, but when both IL-1β and IL-23 are present together, Th17 cell differentiation occurs. In addition to the IL-1β/IL-R1/myeloid differentiation primary response 88 (MyD88) signaling cascade, the activation of MyD88-mediated signaling by the engagement of Toll-like receptors (TLRs) also promotes the differentiation of CD4^+^ T cells toward a Th17 fate [57,58,59,60]. Further, IL-1β and IL-23 alter Th17 metabolism to increase aerobic glycolysis to produce sufficient energy for differentiation, which concordantly supports gene transcription of the Th17 subset [58].

When IL-1β acts together with IL-6, IL-6 upregulates the expression of IL-23 via STAT3, which induces IL-17 gene expression as well as enhancing the expression of RORγt [60]. Additionally, IL-6 upregulates the IL-1R1 receptor expression on Th17 cells, whose signaling promotes expression of transcription factors IRF4 and RORγt [57]. As observed with IL-23, IL-1β also works with IL-6 to alter cell metabolism, thereby allowing for Th17 expansion in inflammatory environments where resources are limited [37].

By comparison, much less is known about the regulation of the expression of other IL-17 family members. The current findings suggest that IL-17C is upregulated by IL-1β, IL-17A, and TNFα, as well as TLR-MyD88 signaling [61,62,63,64]. IL-17C expression is elevated upon activation of p38 MAPK and NF-κB [61,65,66]. IL-17D expression is promoted by transcription factor Nrf2 in response to oxidative stress [5,67]. In certain cell types, IL-17E expression is induced via IL-22 [68].

Studies into cytokine regulation of Th17 cell differentiation have uncovered many important transcriptional pathways. While these may all play specific roles in metabolic diseases and cancers, the RORγt transcription factor should be further investigated, as it is linked to promoting IL-17 induction in other cells as well.

## 3. Transcriptional Control of IL-17 Family Cytokines

The induction of IL-17 cytokines depends on the expression of RORγt transcription factor [46,47,48]. IL-17-producing cells, which will be introduced in the following sections and comprise both innate and adaptive cells, appear to share a dependence on this transcription factor, as well as on receptors to IL-23, IL-1β. They also share the expression of chemokine receptor CCR6 [47,48]. Several additional factors also play a role in inducing IL-17 cytokines through interaction with RORγt. The aryl hydrocarbon receptor (AHR), which is expressed in several IL-17-producing cells, such as those in the gut, skin, and spleen, interacts with RORγt, resulting in an increase in the production of IL-17 cytokines [48]. Additionally, the interaction of RORγt with runt-related transcription factor 1 (RUNX1) in subsets of IL-17-producing cells, such as those in the gut and skin, also results in an increase in the production of IL-17 cytokines, potentially through shifting differentiation toward IL-17-producing cell types [48,49,69]. Importantly, STAT3 is involved in the activation of RORγt upon IL-6 stimulation [48,49]. STAT3, in its active and phosphorylated form, regulates the differentiation and maturation of some subsets of IL-17-producing cells, which can lead to increases in the production of IL-17 when interacting with RORγt [48,50]. Other factors RORγt is known to interact with, resulting in IL-17 family cytokine production, include IRF4, NF-κB, basic leucin zipper transcription factor, ATF-like (BATF), rho-associated kinase (ROCK2), Ets-family transcription factor (Etv5), and sphingosine 1-phosphate-/type 1 S1P receptors (S1P_1S_) [46,70]. In summary, RORγt interacts with a number of transcription factors to induce IL-17 production in a variety of cell types.

## 4. Signaling Events following IL-17 Cytokine–Receptor Engagement

There are several known pathways that can be activated upon the engagement of an IL-17 family receptor by an IL-17 family cytokine. Currently, the known activated signal transduction pathways include MAPK, NF-κB, and C/EBP cascades [17,19,20]. The activation of these pathways involves adaptor proteins, such as Act1, and E3 ubiquitin ligases TRAF2, 5, and 6 [17,19,71]. The activation of intracellular signaling components by IL-17 frequently leads to the production of proinflammatory chemokines and cytokines that are typically associated with innate immune responses [16,17,19,71,72,73]. The following is a brief summary of the signaling events following IL-17 receptor engagement.

The binding of IL-17A or IL-17F homodimers, or the IL-17A/F heterodimer, to IL-17RA and IL-17RC results in the activation of TRAF6-dependent gene transcription [10,74,75]. This ‘canonical’ IL-17 responsive pathway depends on the recruitment of ACT1 adaptor protein that interacts with the SEFIR domain on IL-17 receptors [74,75,76,77,78]. ACT1 interacts with TRAF6 via its TB1 and TB2 domains and facilitates the ubiquitination of TRAF6, leading to TRAF6 autoubiquitination [74,75,76,77]. TRAF6 then ubiquitinates TGFβ-activated kinase 1 (TAK1), which subsequently phosphorylates IKK, leading to IkBα degradation and the activation of NF-κB, as well as MAPK signaling pathways [10,74,75]. In some systems, JAK1-associated PI3K signaling has been shown to mediate NF-κB activation [18,75,79]. Synergy with cytokines such as TNF-α increases the activation of NF-κB [10,18,74,75]. The MAPK signaling pathway has been shown to involve the following transcription modifiers: extracellular signal-regulated kinase (ERK) 1 and 2 (ERK1/ERK2), c-jun N-terminal kinase (JNK), and p38 [2,10,16,71,74,75,80]. Additionally, the C/EBP transcription factor pathway also falls downstream of ACT1 activation [74,81].

Binding of homodimers of IL-17A or IL-17F, or the IL-17A/F heterodimer, to the IL-17RA and IL-17RC receptor complex also increases the half-life of previously transcribed proinflammatory mRNAs via the ‘non-canonical’ signaling pathway [10,74,75,82]. The non-canonical pathway stabilizes these mRNAs through TRAF2- and TRAF5-mediated interactions with molecules controlling mRNA turnover [10,75,83]. This pathway also involves ACT1, but unlike the canonical pathway, it is dependent on TBK1 and IκB kinase (IKKi) phosphorylation of ACT1; it acts as a positive regulator toward the activation of proinflammatory gene transcription pathways [10,74,75,82,83,84].

The binding of IL-17C to the IL-17RA and IL-17RE complex induces signaling via ACT1 to induce NF-κB activation [33]. IL-17C signaling, by this complex, targets and induces IκBζ expression [33]. The recruitment of ACT1 and TRAF6 in IL-17RA and IL-17RB complex activation by IL-17E also results in NF-κB activation [18]. IL-17E also activates ERK, JNK, and p38 in a TRAF6-independent manner, as well as transcription factors JunB, nuclear factor of activated T cells 1 (NFATc1), GATA-binding factor 3 (GATA-3), and STAT6 [18,25,85,86]. Data detailing the IL-17B signaling pathway activated upon interaction with the homodimeric IL-17RB complex are limited. It is known that IL-17B activates NF-κB, which can occur independently of TRAF6 [87]. ERK, JNK, as well as p38 pathways are also activated by IL-17B [87]. IL-17RD has been shown to both regulate IL-17 pathway signaling, as well as negatively regulate NF-κB and IRF signaling pathways downstream of TLRs [31,88,89]. The domains of IL-17RD have been proposed to interact with ERK1/2 MAPK cascade to inhibit fibroblast growth factor signaling [89,90]. When the IL-17RA and IL-17-RD complex is stimulated by IL-17A, IL-17RD sequesters ACT1, disrupting the ACT1/TRAF6 signaling complex, leading to differential regulation of NF-κB and p38 MAPK pathways [89,91,92]. IL-17A signaling through the IL-17RA and IL-17RD complex regulates proinflammatory gene expression in cell-type-dependent manner. IL-17A signal transduction mediated by the IL-17RA and IL-17RD complex competes with IL-17A-induced signaling dependent on TRAF6 recruitment to ACT1, reducing the activation of NF-κB transcription factor and dampening upregulation of IL-6 [31,89,92].

Additional regulatory mechanisms exist in each signaling pathway of the IL-17 family [75]. For instance, inhibition can be achieved when ACT1 interacts with IL-17RD to prevent its interaction with IL-17RA [74]. Ubiquitin-specific peptidase 25 (USP25) and ubiquitin-editing enzyme A20 also deubiquitinate TRAF molecules as means of negative feedback to modulate NF-κB and MAPK activation [74,75]. TRAF3 and TRAF4, likewise, prevent downstream signaling; TRAF3 directly binds IL-17RA, preventing ACT1 recruitment and interaction with TRAF6, whereas TRAF4 competes with TRAF6 through binding at the same location on ACT1 [74]. Moreover, microRNA (miRNA) expression, such as miR-23b, or transcription factors, such as C/EBPδ, can prevent overactivation of proinflammatory pathways [74,75].

The pairing of IL-17 cytokines with their cognate receptors and downstream pathways determines the biological function of these signaling circuits. Further, connecting these IL-17 cytokine/receptor pairs with downstream effector molecules, such as cytokines and chemokines, improves our understanding of how their dysregulation can impact disease.

## 5. Transcriptional Targets of IL-17 Family Cytokines

IL-17 activates the transcription of molecules that are critical for host defense and potentiation of inflammatory responses that are observed in both innate and adaptive arms of the immune system [81,93,94]. While specific pathways activated by IL-17 family receptors upon the binding of an IL-17 family cytokine may differ depending upon cell source, and thereby result in production of different effector molecules, some of the common target genes include proinflammatory and hematopoietic cytokines, chemokines, antimicrobial peptides, and tissue-remodeling substances [74,75,94].

IL-17A is known to induce the expression of several proinflammatory cytokines. A canonical IL-17 target is IL-6, whose production is augmented when IL-17A acts synergistically with TNFα, which also leads to the production of IL-1β and IL-8 [81,94,95,96,97,98]. Additionally, IL-17A acts synergistically with IFNγ to increase the production of IL-8 and monocyte chemoattractant protein-1 (MCP-1) [96,97]. IL-17A also activates the production of IL-17C, IL-36γ, IL-19, and vascular endothelial growth factor (VEGF) [97,99,100]. Chemokines, including CXCL1, CXCL2, CXCL5, CXCL8, CXCL10, CCL2, and CCL20, are also downstream targets of IL-17A [21,81,93,94,95,97,99,100,101,102,103]. In addition, IL-17A induces the production of antimicrobial peptides (AMPs), such as β-defensin, and acute phase proteins, such as lipocalin [81,94]. Moreover, tissue remodeling matrix metalloproteinases (MMPs), which can lead to extracellular matrix destruction and tissue damage, are often induced by IL-17A signaling, notably including MMP1, MMP2, MMP3, MMP9, and MMP13 [98,100,104,105,106,107]. On the other hand, IL-17A downregulates tissue inhibitors of metalloproteinases (TIMP)1 and TIMP2 [106].

IL-17F, which, in addition to being the most homologous cytokine to IL-17A, also exhibits functional similarities with IL-17A. IL-17F stimulates the production of cytokines and chemokines that are proinflammatory, as well as induces tissue-remodeling MMPs and stimulates the production of AMPs [108]. IL-17F induces the production of TNF, IL-1β, IL-6, IL-8, CXCL1, CXCL8, CXCL10, ICAM1, and GM-CSF [21,108,109,110,111,112,113]. IL-17F, although less potently than IL-17A, also induces CCL2, CCL7, TSLP, and MMP13 [108]. MMP1, MMP3, and MCP1 production also results from stimulation with IL-17F [98,113]. Additionally, when combined with TNFα, IL-17F upregulates the expression of G-CSF, IL-6, and MMP3 [4,18,98,108,114]. IL-17F together with IL-22 also results in the production of mhBD2, as well as other antimicrobial peptides [108,115].

While much remains to be investigated regarding the heterodimer of IL-17A/F, this cytokine complex is known to induce proinflammatory molecules, such as IL-6 and CXCL1 [116,117]. IL-17A/F has also been shown to induce the secretion of chemokine CXCL1 [21]. Upregulation of MMP3 as well as podoplanin (PDPN) mRNA expression is also observed upon IL-17A/F stimulation [98]. Additionally, a synergistic effect on the expression of IL-6 and MMP3 mRNA is seen when IL-17A/F is combined with TNF-α [98].

Similarly, IL-17B, IL-17C, and IL-17D can also induce proinflammatory cytokines. IL-17B is known to target cytokine IL-8, as well as chemokines CXCL1, CCL20, and additionally, trefoil factor 1 (TFF1) [118]. Factors upregulated by IL-17B also include antiapoptotic factor BCL-2, stemness factors Oct4, Sox2, Sall4, and the AKT/β-catenin pathway components [119,120]. As with IL-17A, IL-17C induces IL-6, IL-8, and VEGF expression [61,121]. Interestingly, IL-17C also activates the transcription of IL-1β and TNFα [61,121]. Additional proinflammatory cytokines induced by IL-17C include IL-17A/F and IL-22, as well as chemokines CXCL1, CXCL2, CXCL3, and CCL20 [61,64,121]. Further, IL-17C plays a role in upregulating AMPs, such as human β-defensin 2 (hBD2), lipocalin 2 (LCN2), and granzyme B [121]. IL-17C also promotes the expression of anti-apoptotic factors BCL-2 and BCL-X_L_ [61,62]. IL-17D promotes the production of proinflammatory cytokines, including IL-1β, TNFα, IL-6, and IL-8 [5,7,122]. IL-17D also enhances the expression of chemokines CXCL8 and CCL2 [5,122]. Recently, it has been found that IL-17D is important for the production of IL-22 [34]. Additionally, IL-17D targets IL-6 and GM-CSF, and it inhibits hemopoiesis by myeloid progenitor cells [7].

While most IL-17 family cytokines have proinflammatory effects, IL-17E—the least homologous IL-17 family cytokine—is known to induce the production of both pro- and anti-inflammatory cytokines. The cytokine and chemokine targets of IL-17E include IL-1β, TNFα, IL-6, IL-8, MCP-1, CCL5, CCL11, and GM-CSF [8,93,113,123,124,125]. The production of MMP-1 is also stimulated by IL-17E [113]. The synergism of IL-17E with TNFα results in the upregulation of GM-CSF and CXCL8 mRNA, while co-stimulation of IL-17E and TGFβ_1_ upregulates CXCL8 [125]. IL-17E has been found to promote anti-inflammatory effects associated with the expression of IL-4, IL-5, as well as IL-13, and thymic stromal protein (TSLP), while the expression of IL-13 has been associated with suppression of IL-1, IL-6, and IL-23 [93].

While understanding which cytokines IL-17 family members can induce is useful for understanding their general function, the cytokines they induce are strongly influenced by the source cell and their targets. Thus, it is important to provide this biological context to understand IL-17′s role in health and disease.

## 6. IL-17 Producers, Cellular Targets, and Biological Consequences

IL-17 family cytokines execute diverse functions in health and diseases. Such diversity is in part due to the complexity of cellular targets that these cytokines stimulate. Engagement of the same IL-17 ligand on different target cell types can result in the transcription of different genes, leading to tissue-specific functions. In addition, while the targets of IL-17 signaling determine the biological outcome, the sources of IL-17 cytokines also have a heavy hand in determining the immunologic response. Notably, the IL-17 family of cytokines play a vital role in fighting extracellular fungi, viruses, and bacteria—especially at cutaneous and mucosal surfaces. On the other hand, these proinflammatory molecules are also implicated in the pathogenesis of a variety of inflammatory diseases, which will be covered in greater detail later in this review [46,47,93,118,119].

As discussed previously, and shown in Table 1, IL-17A induces a host of pleiotropic effects, as observed from the induction of target genes that encode cytokines, chemokines, antimicrobial peptides, as well as tissue-remodeling substances. There are several sources of IL-17A, which include both innate and adaptive immune cells [126,127,128]. While notoriously associated with the Th17 adaptive immune cell subset, additional cell sources have a hand in producing this cytokine [127,128]. Other adaptive immune cells known to produce IL-17A include CD8^+^ T cells, such as Tc17 cells, as well as resident memory T cells (T_RM_) [126,128,129]. Cells involved in innate defense mechanisms, such as γδT cells, invariant natural killer T cells (iNKT), neutrophils, mucosal-associated invariant T cells (MAIT), as well as type 3 innate lymphoid cells (ILC3), all produce IL-17A [46,126,127,128,130].

The receptor of IL-17 is found on a broad variety of tissues [131]. IL-17A contributes to the production of inflammatory mediators, often by targeting mesenchymal and myeloid cells, and promotes neutrophil-mediated immunity. IL-17 also induces an immune response against microbes. For example, iNKT IL-17A secretion has been found to recruit neutrophils in the respiratory tract [127], whereas in the lung, when secreted by neutrophils, IL-17A promotes the production of IFNγ by T cells [130]. During a bacterial pneumonia infection, IL-17A, produced by CD4^+^ and CD8^+^ T cells, is important for the production of G-CSF and MIP-2, as well as recruiting neutrophils to the site of infection to aid host defense [132].

The amplification of an inflammatory response in psoriatic skin is observed when IL-17A is secreted by neutrophils; however, this point remains controversial due to the phagocytic function of these innate cells [127,133]. Additionally, in the blood and synovia of spondylarthritis and psoriatic arthritis patients, secretion of IL-17A by ILC3s and γδT cells has been found to drive enthesitis and induce the production of cytokines by mesenchymal cells [127]. The inflammatory response in human pancreatic periacinar myofibroblasts due to IL-8 and MCP-1 chemokine secretion is induced by IL-17A from CD4^+^ T cells, which is further exacerbated by IFNγ [96]. Additionally, the expression of IL-1β, TNFα, IL-6, IL-10, IL-2, and PGE_2_ results from IL-17A stimulation on macrophages [72].

As discussed previously, IL-17A also induces the expression of antimicrobial peptides. The production of antimicrobial hBD-2 results from IL-17A stimulation of human airway epithelial cells, which further serves as a chemotaxin to recruit immune cells, such as monocytes, DCs, and memory T cells [134,135]. Innate immunity in the skin is impacted by IL-17A too; the synergism of IL-17A and IL-22 from Th17 cells on keratinocytes results in the expression of AMPs, including β-defensin 2, S100A9, S100A8, and S100A7 [115]. In the gut, where Th17 cells interact with resident gut microbiota, which influence their development, IL-17A induces a protective response against gut microbiota and pathogens through the induction of β-defensins, calprotectin, and lipocalin, as well as other AMPs by intestinal epithelial cells [136,137,138,139,140].

Tissue-remodeling substances can also be expressed as a product of IL-17A stimulation. Treatment with IL-17A on diseased tendon-derived fibroblasts induces MMP3 expression [98]. In the nasal tissue of patients with chronic rhinosinusitis with nasal polyps, CD8^+^ T cell production of IL-17A results in an increase in the expression of MMP-9 by nasal epithelial cells [104].

In lymphocytes, IL-17A secreted by CD4^+^ T cells represses Th1 polarization, but when acting on B cells, it promotes B cell survival, proliferation, and differentiation into plasma cells [94].

IL-17F shares a great amount of overlap with the function and cellular sources of IL-17A, its close homolog. Often inflammatory, IL-17F functions in promoting tissue-mediated innate and adaptive immune responses against microbial infections by bacteria and fungi. IL-17F-producing cells include Th17 cells, LTi, NK cells, iNKT cells, neutrophils, ILC3, and γδ T cells [21,48,141]. The target cells of IL-17F include epithelial cells, endothelial cells, and stromal cells [4,12].

Inflammation induced by IL-17F, as with IL-17A, harbors infiltrate predominantly comprising neutrophils. IL-17F activates the expression of GM-CSF, ICAM-1, IL-6, and IL-8 by bronchial epithelial cells, thus recruiting lymphocytes, neutrophils, and macrophages. This process has been implicated in allergic inflammatory responses in the lung [109,110,111,114]. Additionally, stimulation with IL-17F and TNFα induces chemokine CXCL1 and cytokine G-CSF production in human bronchial epithelial cells, while IL-17F and IL-23 enhance the production of IL-6 and IL-1β from eosinophils linked to allergic inflammation [114,142]. The upregulation of proinflammatory mediators IL-6 and CXCL1 has also been observed, resulting from IL-17F stimulation of fibroblasts and macrophages [116]. In addition to TNFα and IL-23, IL-17F also works with IL-22 to induce the expression of AMPs, such as hBD-2, S100A7, S100A8, and S100A9, from primary keratinocytes as a function of innate immunity in the skin [115].

IL-17A and IL-17F can also form a heterodimer (IL-17A/F) that signals through the same IL-17RA/RC complex [23]. IL-17A/F has been reported to be produced by the same cell types as those producing its homodimer counterparts, such as Th17 cells, γδT cells, and ILCs [18,23,117].

As with IL-17A and IL-17F, IL-17B is involved in fostering an inflammatory response to aid host defense. However, IL-17B signaling is restricted to certain target cells and has been associated with type 2 immunity [6,124,143]. Like IL-17A, IL-17B has been found to be produced by neutrophils. However, unlike IL-17A, IL-17B is not produced by Th17 cells; it is expressed in a wide variety of tissues, albeit not ubiquitously [6,87,124,144,145]. IL-17B is produced by chondrocytes, neurons, stromal cells, intestinal epithelial cells, as well as memory and germinal center B cells [87,124,144,146,147,148]. IL-17B is expressed in the spinal cord, and at lower levels, in the trachea, lung, stomach, small intestine, colon, prostate, testes, ovary, adrenal gland, and pancreas [3,6,144,148]. The known target cells of IL-17B include ILC2s, NKT cells, fibroblasts, and Th2 cells [3,143,145].

Induction of a type 2 immune response by IL-17B is observed when the cytokine acts on peripheral blood mononuclear cells (PBMCs), NKT cells, or ILC2s. Stimulation of PBMCs by IL-17B results in the production of IL-5 and IL-13. Similarly, IL-17B promotes the production of IL-5 by ILC2s [3]. Co-stimulation of NKT cells by IL-17B and IL-25 promotes the release of IL-5 [3]. On the other hand, anti-inflammatory functions of IL-17B are observed through its ability to inhibit the proinflammatory effects of IL-25 in the lung by restraining the Th2 response [149]. Protective effects of IL-17B through type 2 immunity are also observed through its inhibition of IL-6 production by IL-25 in models of colitis [149].

However, a more inflammatory phenotype induced by IL-17B is seen when the cytokine acts on fibroblasts co-stimulated by TNFα, resulting in expression of G-CSF, contributing to neutrophil migration, as well as expression of IL-6 and CCL20, which are Th17 chemotactic and induction factors [145]. Proinflammatory phenotypes are also observed when IL-17B, as well as IL-17C, stimulate THP-1 cells to release TNFα and IL-1β or stimulate 3T3 cell lines and peritoneal exudate cells to release IL-6 and IL-23 [3,150].

Akin to its homologs IL-17A and IL-17B, IL-17C shares a role in host defense by promoting inflammation. IL-17C is heavily involved in the innate immune response against bacteria, fungi, and viruses [61]. IL-17C is predominantly expressed by epithelial cells of the lung, skin, and colon, but it is not found to be expressed by lymphocytes [64,66,124,151,152,153]. Often, IL-17C stimulates an innate response of epithelial cells in an autocrine fashion. For this reason, the inflammation incurred by IL-17C is often linked to the skin [64,152,153]. Additionally, IL-17C acts on Th17 cells to aid their development, thus promoting the production of IL-17A and IL-17F [33,154].

In response to bacterial pathogens and other microbes in the respiratory tract, IL-17C promotes the release of proinflammatory IL-6 from bronchial epithelial cells [152]. Similarly, as a response to intestinal pathogens, IL-17C synergizes with IL-22 to induce the production of AMPs, chemokines, and proinflammatory cytokines from colon epithelial cells, where it is also involved in barrier stability via enhancement of occludin, claudin-1, and claudin-4 expression [155,156]. In mucosal and cutaneous epithelial cells, IL-17C is known to induce proinflammatory IL-1β, as well as TNF, in response to bacterial pathogens [64,153]. Additionally, IL-17C expression is observed in kidney epithelial cells after fungal infection, resulting in expression of proinflammatory cytokines TNFα, IL-1β, as well as IL-6, by renal epithelial cells [157]. IL-17C also has a hand in combating viral infections; when produced by keratinocytes as a response to herpes simplex virus-2, it functions as a neurotrophic factor exerting protective effect on peripheral nerve systems [61,158].

The functional importance of IL-17D has been slowly uncovered but still requires more investigation. Thus far, IL-17D has been implicated in stress response, intestinal homeostasis as well as anti-viral and anti-tumor responses [34,67,159,160]. Expression of IL-17D has been found in fibroblasts, skeletal muscle, brain, adipose tissue, tumor cell lines, heart, lung, pancreas, as well as colon epithelial cells [7,34,67,159,160]. The known target cells of IL-17D include endothelial cells and ILC3s [67]. IL-17D produced by colonic epithelial cells stimulates ILC3 for their production of IL-22 and antimicrobial peptides RegIIIβ and RegIIIγ, thus helping confer protection against bacterial infection, dysbiosis, and colonic inflammation [34]. Additionally, oxidative stress, due to tumor development or viral infection, induces transcription factor Nrf2 to promote IL-17D expression. IL-17D stimulates endothelial cells to express MCP-1 for the recruitment of NK cells to elicit an effective anti-tumor or anti-viral effect [67,159,160]. Further, IL-17D is implicated in inhibiting hemopoiesis and myeloid progenitor proliferation, which is thought to occur through its ability to stimulate the production of cytokines, such as IL-8, from endothelial cells or other target cells [7].

A wider variety of cells produce the pleiotropic IL-17E. IL-17E-producing cells include mast cells, alveolar macrophages, eosinophils, basophils, ILC2s, DCs, stromal and epithelial cells, and Th2 cells [47,68,85,86,124,161,162,163,164,165,166,167]. The target cells of IL-17E include keratinocytes, lung epithelial cells, endothelial cells, Th2 and Th9 cells, macrophages, fibroblasts, and ILC2s [68,163,165,168]. IL-17E protects the host against parasites, such as helminths, as it orchestrates type 2 immunity. IL-17E has also been implicated in allergic and skin inflammation, anti-fungal immunity, inhibition of Th17 development, as well as maintenance of intestinal homeostasis [85,86,151,163,165,167,169,170]. Keratinocyte-produced IL-17E promotes skin inflammation through the recruitment of neutrophils and macrophages [171]. IL-17E also directly targets keratinocytes in an autocrine manner to facilitate wound healing through the induction of keratins 6, 16, and 17. In addition, IL-17E increases the metabolic activity of keratinocytes by activating the mTOR pathway during skin inflammation [68]. Interestingly, IL-17E does not appear to induce the production of β-defensin 2, LL-37, or S100A7 when acting on keratinocytes [68,172,173]. The involvement of IL-17E in antifungal immunity is observed through the ILC2 inflammatory phenotype following IL-17E stimulation, which confers protection against *Candida albicans* [174]. IL-17E also induces Th2 differentiation and promotes the production of Th2 cytokines. For example, IL-17E stimulation of Th9 cells results in the production of IL-9, a cytokine implicated in allergic airway inflammation [168]. Stimulation of Th2 memory cells by IL-17E secreted by basophils and eosinophils, similarly, has been found to promote sustained Th2 transcription factor expression, leading to Th2 expansion and polarization, and allergic inflammation [86]. Additionally, IL-17E recruits macrophages and eosinophils to the lung, promotes Th2 cell differentiation, and stimulates release of Th2 cytokines, such as IL-4, IL-5, and IL-13, and by doing so, it initiates innate and adaptive proallergic responses by lung epithelial cells [85,161]. A type 2 anti-helminth immune response to maintain intestinal homeostasis is imparted through tuft cell-derived IL-17E, which targets ILC2 and Th2 cell-mediated release of IL-4, IL-5, and IL-13 [165,169]. In addition, IL-17E produced by mast cells and keratinocytes acts on dermal DCs to stimulate the production of IL-1β, IL-6, and TNFα, which subsequently activate Th17 cells [68,172,173].

## 7. Pathological Consequences of IL-17 Signaling

Despite their involvement in host protective immune responses, the IL-17 family of cytokines are also implicated in a variety of proinflammatory pathologies. Aberrant expression of the tissue-specific and context-dependent IL-17 family cytokines, or an inappropriate response of target cells to these cytokines, are often associated with autoimmune disorders and cancers. In certain cases, IL-17 family-dependent pathologies, or simply the dysregulation of IL-17 family cytokine production or synergism, are drivers for metabolic disorders. On the other hand, metabolic disorders can increase the severity of IL-17-induced pathology, representing a unique interplay between immunity and metabolism, which will be discussed later in this review.

While the production of IL-17 cytokines during infection occurs in a controlled manner, chronic production of and biological responses to IL-17 can lead to inflammatory disorders, such as psoriasis. Psoriasis is labeled as an IL-17A-driven disease; however, IL-17C, IL-17E as well as IL-17F also contribute to this pathology [63,93,124]. In healthy skin, the presence of IL-17A is beneficial—facilitating epithelial barrier protection against fungi and bacteria, as well as promoting tissue repair and wound healing [93,124]. The host protective responses driven by IL-17A and IL-17F include keratinocyte proliferation and differentiation, production of AMPs and inflammatory and chemoattractant molecules that recruit neutrophils, macrophages, and immune cells, such as Th17 or Tc17 cells [93,124,175]. Additionally, IL-17A and IL-17F can synergize with TNFα to increase the production of inflammatory mediators [63,124,175]. Similarly, IL-17C, which, like IL-17A and IL-17F, is upregulated in psoriatic skin, targets endothelial cells and keratinocytes and works with TNFα to induce inflammation [66,175,176,177]. IL-17E also contributes to this pathology by stimulating keratinocytes and macrophages to produce inflammatory cytokines and chemokines [123,124].

The inflammatory effects of IL-17A are also implicated in bone destruction observed in rheumatoid arthritis (RA) [178]. As seen in psoriasis, the synergy between IL-17A and TNFα is involved in the pathogenesis of RA. IL-17A and TNFα promote inflammation through the production of IL-6 and IL-8 cytokines released from fibroblast-like synoviocytes [95]. Additionally, IL-17A contributes to RA by promoting osteoclast differentiation and inducing chemokines that recruit neutrophils, macrophages, as well as other lymphocytes to the joint, eventually leading to the destruction of synovium [178,179,180,181]. Moreover, Th17 cells not only contribute to the inflammatory milieu observed in RA, but also induce MMPs that further the destruction of synovial tissues [179,180,181].

Inflammatory bowel disease (IBD) is another autoimmune condition where the IL-17 family of cytokines are involved. IBD encompasses both ulcerative colitis (UC) and Crohn’s disease (CD). Although complex, as Th17 cells function in maintaining gut homeostasis, the number of Th17 cells and the levels of their associated cytokines are increased in intestinal mucosa of IBD patients [182,183]. IL-17A, IL-17C, and IL-17F are all overexpressed in IBD [184,185,186]. Specifically, IL-17A is responsible for amplifying TNFα-induced IL-17C production by enteroendocrine and goblet cells. IL-17C subsequently upregulates CCL20 to allow for further influx of Th17 cells [186]. Increased expression of CXCL1 in CD and CXCL6 in UC by intestinal fibroblasts following IL-17A stimulation also contributes to IBD pathology [187].

Additional disorders are also linked to the IL-17 family cytokines. To highlight a few: asthma, Behçet’s disease, multiple sclerosis, systemic lupus erythematosus, ankylosing spondylarthritis, systemic sclerosis, and uveitis [128,188,189,190,191,192]. While IL-17 is linked to a variety of inflammatory disorders, more research into how IL-17 influences these disorders may uncover novel therapeutic targets.

## 8. IL-17 and Cancer—A Dangerous Liaison

The IL-17 family of cytokines have been associated with multiple types of cancers. IL-17A, IL-17B, IL-17C, as well as IL-17F have been shown to promote tumor development, whereas IL-17D and IL-17E play anti-cancer roles in a context-dependent manner by activating the Nrf2 stress surveillance pathway, recruiting innate immune cells, and activating leukocytes [67,159,193,194,195].

Breast cancer growth, migration, angiogenesis, and invasiveness are further enhanced by IL-17A [196,197,198]. IL-17A promotes ERK1/2 phosphorylation and p38 MAPK activation, which in turn promotes breast cancer cell proliferation, migration, and tissue invasion [196,197,198,199]. Additionally, IL-17A alters the tumor microenvironment through the induction of tissue-remodeling substances, such as MMPs, inhibition of apoptosis, promotion of angiogenesis as well as recruitment of immature myeloid cells, which can suppress the activity of CD8^+^ T cells [198,199,200,201,202]. While IL-17E has been shown to induce apoptosis in breast cancer cells, IL-17B as well as IL-17RB expression correlates with worse cancer outcomes [120,203,204]. Elevated levels of IL-17B/IL-17RB signaling is associated with enhanced tumorigenicity due to ERK1/2 pathway activation leading to upregulation of Bcl-2 expression, as well as HER2 amplification [120,203].

IL-17A and IL-17C both promote the development of colorectal cancer (CRC) [93,205,206]. IL-17A promotes CRC cell proliferation, tumor growth, and angiogenesis resulting from VEGF production in CRC. IL-17C, in response to stimulation by gut microbes, upregulates the expression of Bcl-2 and Bcl-x_L_ in intestinal epithelial cells, promoting cell survival [62,202,207]. Moreover, IL-17C promotes tumor angiogenesis in CRC [208]. IL-17F is another player in CRC; in different contexts, IL-17F exerts a protective anti-angiogenic effect or pro-tumorigenic effect through promotion of the epithelial–mesenchymal transition [195,209,210]. The role of IL-17B in CRC requires additional investigation, but its overexpression could be linked to increased inflammatory cytokine expression [211]. A review by Razi et al. provides further detail on IL-17A in CRC [212]. The context-dependent malignant versus protective role of IL-17F is seen in a variety of cancer types and has been extensively investigated by Mikkola et al. [195].

Although not exhaustive, the IL-17 family of cytokines have also been implicated in cancers of the lung, prostate, cervix, ovary, bladder, and skin [213,214,215,216,217,218,219,220,221,222]. These areas have been extensively covered by a number of review articles and will not be elaborated here. Nonetheless, the tumor-promoting roles of IL-17 family cytokines share common mechanistic traits, such as promoting myeloid cell infiltration, enhancing cancer cell survival and growth, and inhibiting anti-cancer immune responses (Figure 1).

Tumor-infiltrating dendritic cells (DC) and other myeloid cells respond to stimuli present in the tumor microenvironment and produce multiple cytokines to drive the differentiation and activation of IL-17-producing Th17 cells and other immune cells. IL-17, in turn, signals to tumor cells and turns on genes that are responsible for tumor cell survival and proliferation, as well as factors that further exacerbate tumor-associated inflammation and angiogenesis. IL-17 also inhibits the activity of anti-tumor immune cells by recruiting myeloid-derived suppressor cells (MDSC) and inhibiting the production of T-cell-attracting chemokines. Overall, hijacking IL-17 signaling by tumors results in a pro-tumor inflammatory milieu that favors tumor growth and impedes tumor destruction.

## 9. IL-17 Family Cytokines That Fuel Metabolic Disorders

The IL-17 family of cytokines are involved in, and influence, metabolic processes. IL-17A has been shown to induce differential expression of genes in pre-adipocytes versus adipocytes [223]. In murine models, IL-17A negatively regulates adipogenesis and glucose metabolism [223,224]. This is mediated by IL-17′s function in downregulating genes that are important for both lipid and glucose metabolism [224,225]. In human bone marrow mesenchymal stem cells (hBM-MSC), IL-17A increases the expression of IL-17RC, which is associated with inhibition of adipocyte differentiation [225]. However, in differentiated adipocytes, IL-17A inhibits adipose tissue accumulation by preventing glucose uptake [224]. In fully differentiated adipocytes, IL-17A decreases the transcriptional activities of peroxisome proliferator activating receptor γ (PPARγ) and fatty acid binding protein 4 (FABP4), increases lipolysis, stimulates IL-6, TNFα, and IL-8 cytokine production, and upregulates COX2 gene expression, which consequently increases PGE_2_ levels [225,226]. Interestingly, previous reports have demonstrated the immunomodulatory effects of PGE_2_ produced by MSCs on fully differentiated Th17 cell cytokine production, where PGE_2_ production by MSCs is augmented upon cell–cell contact with Th17 cells, likely due to the presence of TNFα secreted by Th17 cells [227]. TNFα has been cited as important in lipid homeostasis, acting as an adipokine produced from adipose tissue that is important in inducing insulin resistance, lipolysis, and inhibiting adipocyte differentiation [228].

MSCs further suppress adipogenesis by inhibiting the differentiation of naïve T cells into Th17 cells [227]. In 3T3-L1 cells, IL-17A-mediated suppression of adipogenesis is observed through suppression of Kruppel-like family transcription factor (KLF) 15, PPARγ, as well as C/EBPα [229]. However, such effect was not observed in the adipose tissue of obese individuals, warranting further study to elucidate the context-dependent regulation of adipogenesis by IL-17 [230]. Of note, there is a high level of IL-6 expression in undifferentiated MSCs, which is necessary for undifferentiated MSC proliferation, as well as inhibition of the expression of FABP4 and LPL genes [226]. In MSCs, IL-6 activates the ERK1/2 pathway [226]. However, the overall contribution of IL-6 to metabolism is complex. Recent reports describe the contribution of IL-6 in metabolism as dependent upon the source, signaling pathway—canonical or noncanonical—and diet, which subsequently influences leukocyte accumulation and inflammation [231]. IL-6 secreted from adipocytes is also associated with adipose tissue macrophage accumulation [231].

In adipose tissue, leukocytes, such as γδ T cells, have been identified as the main IL-17A producers [224,232]. Recently, γδ T cells, which were previously linked to controlling body temperature via IL-17A expression, have been shown to promote adaptive thermogenesis through sympathetic innervation resulting from stimulation of IL-17F binding IL-17RC, driving parenchymal cell production of TGFβ1 in adipocytes [232].

Certain metabolic disorders have been linked with the IL-17 family of cytokines and their dysregulation. Alterations in the levels of cytokines important in the expression of the IL-17 family of cytokines or their targets have also been implicated. For example, IL-17A has been associated and studied for its contribution to obesity [223,230,233,234,235,236]. In investigating the differences relating to IL-17A observed between obese mice that were fed a high-fat diet (HFD) with lean mice on a regular diet (RD), Qu et al. observed elevated levels of IL-17A in subcutaneous adipose tissue (SAT) of obese mice [223]. Congruently, in a separate study, adipose tissue from obese human subjects was found to promote IL-17A release from CD4^+^ T cells, which was positively correlated with IL-1β concentrations in conditioned media from omental adipose tissue [233]. Additionally, Qu et al. reported that IL-17A significantly increases inflammatory cytokines in adipose tissue of obese mice. However, lean mouse visceral adipose tissue (VAT) expresses a greater level of IL-6 at baseline [223,230]. Human pre-adipocytes, in the presence of IL-17A or IL-1β, upregulate a host of genes related to fibrosis and inflammation [233]. In human pre-adipocytes, IL-17A, to a lesser extent than IL-1β, upregulates MMP1, MMP3, MMP8, IL-1β, CCL2, CEBPB, SERPINE1, and GREM1. In human adipocytes, IL-6, IL-8, CCL2, and CCL20 are induced by IL-17A, again, to a lesser extent than IL-1β [233]. Additionally, the differences in metabolic gene expression as well as signaling pathway activation in obese mouse adipose tissue versus lean mouse adipose tissue reveal that IL-17A in obese mice induces expression of genes associated with diet-induced obesity (DIO) and dampen glucose transport [223]. Concordantly, alterations in metabolic genes in human adipocytes were associated with IL-17A and IL-1β signaling on adipocytes [233]. Moreover, IL-17A increases STAT3 signaling in obese mouse adipose tissue [223].

Prevention of DIO and the associated metabolic and inflammatory disturbances is seen upon suppression of IL-17A expression via inhibition of RORγt by ubiquitous deletion of IL-17RA or use of IL-17A and IL-1β neutralizing antibodies in adipocytes [233,236]. Global inhibition of IL-17RA is associated with thermogenesis, as well as decreased leukocyte infiltration into adipose tissue [236]. Further, reduced obesity and improved metabolic markers are observed by IL-17RA deletion from adipocytes, where IL-17A phosphorylation of PPARγ is associated with the repression of browning and expression of diabetogenic and obese genes [236]. Further, IL-17A deficiency inhibits TANK-binding kinase 1 (TBK1) activation in adipocytes and is also associated with reduced inflammation in obese mice [234].

In addition to increased inflammation and alterations in metabolic markers, obesity is associated with other IL-17 family cytokine-related diseases. Obesity has been identified as a risk factor for psoriasis, and likewise, a trend in weight increase in patients with psoriasis has been observed [237,238,239]. The attribution of the bidirectional risk is speculated due to the overlap of inflammatory cytokines involved in both diseases, cellular sources of inflammatory cytokines, and alterations in oxidative stress levels [240,241]. A bidirectional association and risk of obesity are also observed with rheumatoid arthritis, which is similarly attributed to increased inflammatory cytokine levels, as well as leukocyte recruitment [242,243]. In addition, the altered abundances of metabolites in obesity and other metabolic disorders may function as immune regulators independent from their conventional role in metabolism [244]. It is also important to note that inflammation does not necessarily stem from host obesity status. For example, a diet high in fat, such as the standard Western diet, has been linked to inflammation due to the alteration in gut microbiota causing dysbiosis, barrier dysfunction, and inflammation [245].

Beyond, and including obesity, metabolic dysregulation linked with the IL-17 family of cytokines has been associated with certain cancers. A mechanistic consideration linking the involvement of the IL-17 family of cytokines with metabolic dysregulation leading to cancer will be discussed in the next section of this review.

## 10. IL-17 Bridges Metabolism and Cancer

Increasing evidence suggests that dysregulated metabolic pathways accompany the process of tumorigenesis. Metabolic syndrome (MetS), which is clinically characterized by the presence of at least three metabolic abnormalities, including insulin resistance and high fasting blood sugar levels, dyslipidemia and low fasting HDL cholesterol, obesity with an emphasis on abdominal obesity, and hypertension, has been clinically and experimentally associated as a risk factor for a variety of cancers and associated with a worse clinical outcome [246,247,248,249,250,251,252]. A meta-analysis has revealed gender- and population-specific differences in risks associated with developing certain cancer types in those with MetS [247]. MetS is associated with increased risks of liver and colorectal cancer in males, and pancreatic and rectal cancer in females [247]. The presence of MetS is also associated with breast cancer risk, as well as with total cancer mortality [248,249,250,251,252]. Individual components that characterize MetS have been associated with total cancer mortality risk and all-cause cancer mortality in men [249,252]. Although it is well established that MetS is associated with an increased risk of cancer, the mechanisms linking both pathologies require further investigation. When examining individual components of MetS, the IL-17 family of cytokines may provide a link between metabolic dysregulation and the path to developing cancer.

Recent advances in cancer biology point to a role of IL-17 in linking metabolic disorders and cancers (Figure 2). γδ T cells that produce IFN-γ and IL-17 exhibit distinct metabolic preferences [253]. While IFN-γ-producing γδ T cells rely on glycolysis, IL-17-producing γδ T cells undergo predominantly oxidative metabolism [253]. This selective preference of metabolism was imprinted in the thymus during T-cell development and maintained in the periphery. Tumor-promoting IL-17^+^ γδ T cells showed high lipid uptake and storage and were expanded in tumors of obese mice [253]. Therefore, obesity contributes to tumor development by upregulating the production of IL-17 by γδ T cells. On the contrary, IFNγ-secreting γδ T cells were associated with tumor regression and thrived in a glucose-rich environment [253,254]. In separate studies examining the impact of obesity on T-cell antiviral immunity, obesity was found to negatively impact the percentage of Vγ9Vδ2 T cells, an IFNγ-secreting γδ T-cell subset [255,256].

Recent studies on IL-17 have started to link its function with both metabolic disorders and cancer. IL-17 may promote tumor development following disrupted metabolism in one of the following three ways: (1) obesity enhances tumor development by promoting IL-17 production from γδ T cells within the tumor environment. At the same time, the production of anti-tumor IFN-γ by γδ T cells is reduced; (2) hypercholesteremia induces higher levels of IL-17 in the serum and promotes tumor growth; (3) obesity disrupts IL-17-mediated regulation on intestinal microbiota, which results in glucose intolerance and insulin resistance, which are also linked to increased risk in cancer development.

Hypercholesteremia, a metabolic abnormality and component of MetS, where serum levels of low-density lipoprotein cholesterol (LDL-C) are elevated, is associated with increased risk of atherosclerotic cardiovascular disease as well as cancer [257,258]. Interestingly, in a study by Roohi et al. involving serum collected from 120 healthy subjects, a statistically significant difference in serum IL-17A levels was found between those with LDL-C ≥ 130 mg/dL when compared with those with LDL-C < 130 mg/dL [259]. The positive association of serum IL-17A with LDL-C through the perturbation of IL-17A levels, as well as IL-23 levels, may pave the road for the development of MetS and cancer. In white males, low levels of high-density lipoprotein (HDL) and high levels of LDL are associated with an increased risk of prostate cancer [260]. Experimentally, LDL-C was found to induce phenotypic and genotypic changes in ER-positive breast cancer cells [261]. In vitro LDL-C was found to induce ER-positive breast cancer cell proliferation and reduce cell adhesion, which results from upregulation of genes related to cell survival and proliferation pathways as well as downregulation of adhesion molecules [261]. Additionally, in vivo, LDL-C is found to promote ER-positive breast cancer growth [261]. Hypercholesterolemia is also associated with increased incidence of CRC [262]. In a study by Tie et al., the presence of hypercholesteremia in mice increased the number of azoxymethane-induced colorectal tumors [262]. The authors found reduced differentiation of γδT and NKT thymic and colon submucosa populations via an increase in oxidative stress on hematopoietic stem cells (HSCs) due to oxidized LDL [262]. The skewing of lineage fate, controlled by downregulation of Tet1, also impacts the function of terminally differentiated γδT and NKT cells [262]. In ApoE^-/-^ mice, γδT cells were found to produce less IL-17. However, γδT cells derived from Tet1-overexpressing HSCs produced greater amounts of IL-17 when compared with wild-type mice [262]. However, it is unknown whether the change in IL-17 level is responsible for the altered rate of CRC development.

Another way in which the IL-17 cytokine family may link metabolic dysregulation with cancer is through its impact on intestinal microbiota [263,264]. Th17 cells in the small intestine have been found to promote the expansion of gut microbiota associated with leanness and metabolic homeostasis in an IL-17-dependent manner [263]. HFD-fed obese mice have reduced Th17 cells in the small intestine as well as increased IFNγ^+^ Th1 cells and γδ T cells [263]. Similarly, in patients with MetS, serum levels of IL-17 are reduced in conjunction with the increasing presence of metabolic abnormalities [265]. IL-17 signaling to intestinal epithelial cells drives changes in the microbiota, and this mechanism is required for the resistance to metabolic disorders induced by a HFD [264]. This intestinal-specific IL-17–microbiota–metabolism axis is of special interest because it differs from the traditional view of obesity as a proinflammatory state, where researchers found increased levels of IL-17A in the serum [266,267] and visceral adipose tissues [268]. There has already been evidence linking disrupted glucose metabolism with cancer. Elevated levels of blood glucose, hyperinsulinemia, and insulin resistance are associated with increased risk of CRC mortality in both sexes [269]. Insulin resistance in postmenopausal women also increases both risk of and death from breast cancer [270,271].

## 11. Future Perspectives

The development of effective anticancer therapeutics and immunotherapies is exceptionally complex. Issues range from combating drug toxicity and adverse side effects, harnessing a drug or patient’s immune system to target malignant tumor cells effectively and selectively, to molecular resistance observed through the course of treatment. Additionally, the complications are compounded by the presence of metabolic diseases or when considering the bidirectional impact of anticancer therapies on metabolism.

As discussed, alterations in metabolism can support the development of tumors. The distinct metabolic demands of cancer cells have been harnessed in cancer therapy, with most research uncovering vulnerabilities of malignant and rapidly proliferating cancer cells. Additional research has been poured into understanding lymphocyte metabolism in the tumor microenvironment to further harness drug specificity and therapeutic potential.

IL-17, as a family of proinflammatory cytokines, has been implicated in both metabolic disorders as well as cancer development and resistance to therapy. Further investigation on the role of IL-17 as a link between disrupted metabolism and cancer is needed to elucidate the function and mechanism of action of such link, so as to validate the potential of the IL-17 pathway as a target for novel cancer therapy in humans. Pharmacological agents against different members of IL-17 and their receptors have been proven effective and safe for the treatment of autoimmune diseases in humans. Notable ones include Secukinumab and Ixekizumab targeting IL-17A, Bimekizumab targeting IL-17A and F, and Brodalumab targeting IL-17RA [272]. Inhibitors of IL-17E are also gaining traction for the treatment of asthma, and more recently, psoriasis [123,273,274]. These agents could be tested in a trial on IL-17 pathway blockade as adjuvant therapy against cancers in humans, especially those who also suffer from metabolic syndromes.

## Figures and Tables

**Figure 1 genes-13-01643-f001:**
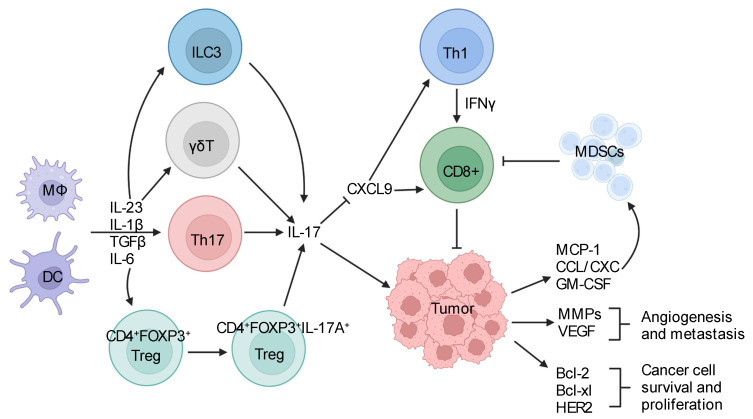
How IL-17 promotes cancer development.

**Figure 2 genes-13-01643-f002:**
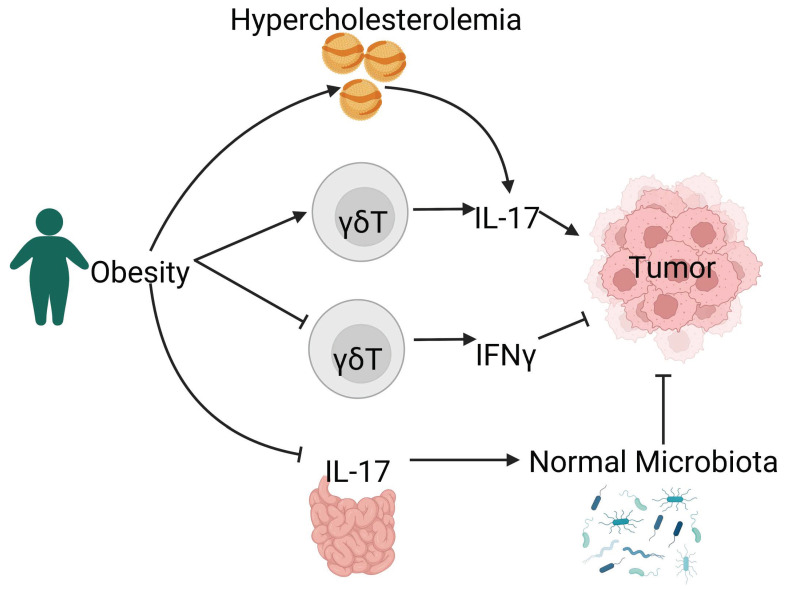
IL-17 bridges metabolic disorders and cancer.

**Table 1 genes-13-01643-t001:** IL-17 cytokine family producing cells, targets, and effector molecules.

Cytokine	Cellular Sources	Target Cells	Downstream Effector Molecules	References
IL-17A	Th17, Tc17, T_RM_, γδT, iNKT, neutrophils, MAIT, ILC3	Mesenchymal cells, myeloid cells, macrophages, epithelial cells, keratinocytes, intestinal and nasal epithelial cells, B cells, CD4^+^ T cells	G-CSF, MIP2, IL-8, MCP-1, IL-1β, TNFα, IL-6, Il-10, IL-2, PGE_2_, hBD-2, β-defensin, S100A9, S100A8, S100A7, calprotectin, lipocalin, MMP-9	[46,72,94,96,104,115,126,127,128,132,134,135,136,137,138,139,140]
IL-17B	Neutrophils, chondrocytes, neurons, stromal cells, gut epithelium, B cells (germinal and memory)	ILC2, NKT, THP-1 monocytes, fibroblasts, Th2, PBMC, ILC2	IL-5, IL-13, G-CSF, IL-6, CCL20, TNFα, IL-1β	[3,87,124,143,144,145,146,147,148]
IL-17C	Epithelial cells	Epithelial cells, Th17	G-CSF, IL-6, IL-1β, TNFα, AMPs, occluding, claudin-1, claudin-4	[64,66,124,151,152,153,155,156,157]
IL-17D	Fibroblasts, colonic epithelial cells, brain, skeletal muscle, adipose tissue, heart, lung, pancreas	Endothelial cells, ILC3	IL-22, RegIIIβ/γ, MCP-1, IL-8, IL-6, GM-CSF	[7,34,67,159,160]
IL-17E (IL-25)	Mast cells, alveolar macrophages, eosinophils, basophils, ILC2, dendritic cells, stromal cell, epithelial cells, Th2	Keratinocytes, lung epithelial cells, endothelial cells, Th2, Th9, macrophages, fibroblasts, ILC2, DCs	Keratin 6, keratin 16, keratin 17, IL-1β, IL-9, IL-4, IL-5, IL-6, IL-13, TNFα	[47,68,85,86,124,161,162,163,164,165,166,167,172,173]
IL-17F	Th17, LTi, NK, iNKT, neutrophils, ILC3, γδT	Epithelial cells, endothelial cells, stromal cells, eosinophils, fibroblasts, macrophages, primary keratinocytes	CXCL1, G-CSF, GM-CSF, ICAM-1, IL-6, IL-8, IL-1β, CXCL, hBD-2, S100A7, S100A8, S100A9	[4,12,21,48,109,110,111,114,115,116,141,142]
IL-17A/F	Th17, γδT, iNKT, NKp46+	Epithelial cells, fibroblasts	CXCL1, IL-6, IL-8	[18,23,117]

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
