# Peer review of "Interleukin-17 Family Cytokines in Metabolic Disorders and Cancer"

_genes, 2022, doi:10.3390/genes13091643_

Round 1

Reviewer 1 Report

Eileen Victoria Meehan , Kepeng Wang “Interleukin-17 family cytokines in metabolic disorders and cancer “

This is a timely and comprehensive review on the subject which covers some of well described roles of IL-17, capitalizes on recent knowledge in cancer and adds a metabolic spin which is not often mentioned/described for this family of cytokines.

I only have few comments  on how to improve the review:

1)” sequence similarity” for IL17 members- please specify- nucleic acid or protein? Sometimes it says amino acid, but sometimes- not.

2) “review by Zhang et. al (2011)” is this review still current and most detailed?

3)” Multiple cytokine pathways guide the differentiation of IL-17 producing cells”-  seems to be centered around IL-17A and IL-17A producing cells (the same goes to the next section on Transcriptional control). What is known (similar or different) about other IL-17-  F, D, E, C?

4) IL-17 in metabolic disorders….Adipocytes and adipose tissue are nicely described, maybe it is worthwhile to add some info on IL-17 signaling in the liver and muscles or other tissues?

5) Fig 2 -  AUTHORS propose here that Obesity is important or dietary changes which lead to obesity and other issues?  Diet could also influence barrier function and changes in inflammation and regulation of host-microbiota interactions

Author Response

We thank the reviewers for their enthusiastic comments and constructive advices. We have made the following revisions accordingly.

1)” sequence similarity” for IL17 members- please specify- nucleic acid or protein? Sometimes it says amino acid, but sometimes- not.

All sequence similarities were based on amino acid sequences. This point is clarified in the text. (Lines 24 and 26).

2) “review by Zhang et. al (2011)” is this review still current and most detailed?

The review article by Zhang et al is a nice summary on the structural basis of IL-17 family signaling. We also cited a more recent article on the topic by Liu (2019). The change is on line 35.

3)” Multiple cytokine pathways guide the differentiation of IL-17 producing cells”-  seems to be centered around IL-17A and IL-17A producing cells (the same goes to the next section on Transcriptional control). What is known (similar or different) about other IL-17-  F, D, E, C? 

Less is known about the regulation of IL-17 family members other than IL-17A and F. We added a paragraph commenting on this area in lines 156-161.

4) IL-17 in metabolic disorders….Adipocytes and adipose tissue are nicely described, maybe it is worthwhile to add some info on IL-17 signaling in the liver and muscles or other tissues?

We thank the reviewer for the comments. We did not find enough evidence showing IL-17’s role in regulating metabolism of liver and skeletal muscle in health or metabolic disorders. IL-17 is known to promote liver steatosis during fatty liver disease and alcohol consumption, but the impact on systemic metabolism is not clear. We decided to leave this part out until further evidence is available.

5) Fig 2 -  AUTHORS propose here that Obesity is important or dietary changes which lead to obesity and other issues?  Diet could also influence barrier function and changes in inflammation and regulation of host-microbiota interactions

This is great point. We thank the Reviewer for point out this. We added a paragraph introducing to diet altered barrier function and inflammation that is independent of obesity in lines 668-671.

Reviewer 2 Report

The manuscript shed a light on various aspects of IL-17 family cytokines involvement in cellular, immune, genetic and posttranslational processes. The authors describe classification of IL-17 family subunits and the receptors with emphasis of possible interaction among subfamily members/receptors. They also reveals the peculiarities of differental tissue expression of IL17 subfamily cytokines, describe cytokine cascade activation during the action of IL17 family cytokines as well the involvment of IL17 family members in different disease pathogenesis, paying most attention to cancer and metabolic syndrome. The issue discussed in the manuscript is on the top of interest for clinical, molecular immunology, immunogenetics and cancer pharmacology specialists.

Despite excellent scietific content of the manuscript I would like the authors to make a few corrections in the text. It is desirable to note an unusual properties of IL-17E 'Unlike IL-17, IL-25 is not capable of inducing antimicrobial peptides, beta-defensin and LL-37 in keratinocytes', '...dermal dendritic cells release IL-1b in responce to IL-25, and IL-1b directly activates Th17 cells' described by Hasegawa et al. (2022). Besides, IL-17E selective inhibitors are one of the new perspective 'biologics' anti-psoriatic drugs, and this fact maybe mentioned in the review. Regarding the interrelation of interleukin 17 family members on metabolic disorders, I recommend considering the issue in the light of immunometabolomics (eg., Zaslona &O'Neill, 2020).

Author Response

We thank the reviewers for their enthusiastic comments and constructive advices. We have made the following revisions accordingly.

Despite excellent scietific content of the manuscript I would like the authors to make a few corrections in the text. It is desirable to note an unusual properties of IL-17E 'Unlike IL-17, IL-25 is not capable of inducing antimicrobial peptides, beta-defensin and LL-37 in keratinocytes', '...dermal dendritic cells release IL-1b in responce to IL-25, and IL-1b directly activates Th17 cells' described by Hasegawa et al. (2022). Besides, IL-17E selective inhibitors are one of the new perspective 'biologics' anti-psoriatic drugs, and this fact maybe mentioned in the review. Regarding the interrelation of interleukin 17 family members on metabolic disorders, I recommend considering the issue in the light of immunometabolomics (eg., Zaslona &O'Neill, 2020).

We thank the Reviewer for the constructive comments. We added the information on IL-17E’s function in keratinocytes (lines 473-474, 487-490), and cited the Hasegawa paper and additional original research articles.

We added IL-17E selective inhibitors to the last part of the review (lines 778-779).

We also added citation to the immune metabolite paper by Zaslona and O’Neill 2020 to lines 666-668.